# Alleviation of Plant Abiotic Stress: Mechanistic Insights into Emerging Applications of Phosphate-Solubilizing Microorganisms in Agriculture

**DOI:** 10.3390/plants14101558

**Published:** 2025-05-21

**Authors:** Xiujie Wang, Zhe Li, Qi Li, Zhenqi Hu

**Affiliations:** 1School of Environment Science and Spatial Informatics, China University of Mining and Technology, Xuzhou 221116, China; 2School of Plant and Environmental Sciences, Virginia Tech, Blacksburg, VA 24061, USA

**Keywords:** phosphate-solubilizing microorganisms, plant stress tolerance, heavy metal, drought, salinity stress

## Abstract

Global agricultural productivity and ecosystem sustainability face escalating threats from multiple abiotic stresses, particularly heavy metal contamination, drought, and soil salinization. In this context, developing effective strategies to enhance plant stress tolerance has emerged as a critical research frontier. Phosphate-solubilizing microorganisms (PSMs) have garnered significant scientific attention due to their capacity to convert insoluble soil phosphorus into plant-available forms through metabolite production, and concurrently exhibiting multifaceted plant growth-promoting traits. Notably, PSMs demonstrate remarkable potential in enhancing plant resilience and productivity under multiple stress conditions. This review article systematically examines current applications of PSMs in typical abiotic stress environments, including heavy metal-polluted soils, arid ecosystems, and saline–alkaline lands. We comprehensively analyze the stress-alleviation effects of PSMs and elucidate their underlying mechanisms. Furthermore, we identify key knowledge gaps and propose future research directions in microbial-assisted phytoremediation and stress-mitigation strategies, offering novel insights for developing next-generation bioinoculants and advancing sustainable agricultural practices in challenging environments.

## 1. Introduction

Global agricultural systems are confronting intensifying threats from a suite of soil degradation processes—notably contamination, salinization, and drought—which are synergistically amplified by unsustainable anthropogenic practices and accelerating climate change dynamics [1,2]. Soil contamination primarily originates from industrial wastewater discharge, mining activities, and excessive agrochemical use, leading to the accumulation of heavy metals such as cadmium (Cd), chromium (Cr), lead (Pb), copper (Cu), zinc (Zn), and mercury (Hg) [3]. Concurrently, drought and saline–alkaline stress have emerged as critical constraints to crop production. Climate change-driven aridification has expanded drought-affected areas, while seawater intrusion and poor agricultural practices have increased saline–alkaline land coverage [4]. These stressors degrade soil properties, impair microbial community functions, reduce fertility, and destabilize soil ecosystems. Traditional soil-remediation strategies—such as physical adsorption, ion exchange, and electrochemical treatments for heavy metals—partially mitigate contamination but face limitations in cost, operational complexity, and secondary pollution risks [5,6]. Similarly, conventional approaches to saline–alkaline soils (e.g., chemical amendments) and drought (e.g., extensive irrigation) are constrained by environmental and economic feasibility [7]. In contrast, enhancing plant stress resilience through biological strategies offers a sustainable alternative [8].

Phosphate-solubilizing microorganisms (PSMs), such as phosphate-solubilizing bacteria (PSB), fungi, and actinomycetes, are functional microbes capable of converting insoluble soil P into plant-available forms, playing critical roles in the biogeochemical cycling of phosphorus in natural ecosystems [9,10,11]. Beyond P mobilization, many PSM strains synthesize phytohormones, which promote root development and nutrient uptake under stress [12]. Given the pivotal role of PSMs in enhancing soil fertility and plant growth, extensive research has been devoted to their development as biofertilizers, with widespread adoption in modern agriculture [8,13]. Emerging evidence reveals their dual function in stress mitigation: while solubilized P improves soil fertility, microbial metabolites concurrently activate plant antioxidant systems, alleviating oxidative damage from abiotic stressors such as heavy metals, drought, and saline–alkaline conditions [14,15].

A keyword co-occurrence network based on 261 related articles in recent five years (2021–2025) reveals an interdisciplinary research landscape bridging soil chemistry, soil microbiology, and plant physiology (Figure 1). Core terms such as agriculture, soil, and plant delineate the primary application domains, while the strong linkages between phosphate-solubilizing bacteria (PSB) and growth promotion or nutrient uptake underscore their pivotal role in agroecosystems. Notably, the close associations of PSB with heavy metals, drought stress, and salinity highlight their emerging significance in enhancing plant stress resilience, further evidenced by connections to nodes like tolerance and stress response. Emerging keywords (e.g., 1-aminocyclopropane-1-carboxylate (ACC) deaminase, gene expression) reflect a paradigm shift toward elucidating mechanistic pathways and bioremediation processes [16,17,18,19,20]. For instance, Alemneh et al. (2021) demonstrated a synergistic relationship between PSB-mediated phosphate solubilization and ACC deaminase activity [16]. When ACC served as the sole nitrogen source, microbial ACC deaminase production correlated significantly with solubilized phosphorus release from calcium phosphate (Ca-P), a process modulated by medium acidification [21]. Such findings not only advance sustainable agricultural practices but also provide critical insights for future research in microbial biotechnology and environmental stewardship. However, comprehensive reviews addressing PSMs’ applications across multifactorial stresses remain limited. To bridge this gap, our study systematically examines PSM-mediated stress-tolerance mechanisms, highlights persistent challenges, and outlines future research priorities to harness their full potential in sustainable agriculture.

## 2. Roles and Multifunctional Traits of PSMs in Soil

Phosphate-solubilizing microorganisms (PSMs) are found in various environments, including soil [22,23,24], plant root [25,26], sediments [27,28], water [29], and even air [30]. PSMs encompass a vast diversity in nature, with bacteria predominating (1–50% of soil bacteria) over fungi (0.1–0.5%) [31,32,33]. To study and utilize these PSMs, researchers use specific culture media to cultivate them from nature. For phosphate-solubilizing bacteria (PSB), common media include Pikovskaya (PVK) and National Botanical Research Institute phosphate (NBRIP) media [34,35]. For phosphate-solubilizing fungi (PSF), media like Czapek Yeast Extract Agar and Potato Dextrose Agar are often used [36,37,38]. A wide range of PSMs have been identified, such as the bacteria Bacillus, Pseudomonas, and Rhizobium, and the fungi Penicillium, Aspergillus, Trichoderma, Streptomyces, and Streptoverticillium [39,40,41], and new PSM strains have been keeping reported. For instance, Doilom et al. [30] isolated and identified an airborne fungal (*Aspergillus hydei* sp. *nov*.) with significant phosphate-solubilizing activity, which can potentially support plant growth in practical applications. However, it was estimated that more than 60% of PSMs were non-culturable using conventional media [42,43].

The abundance and community diversity of culturable PSMs are profoundly regulated by environmental conditions. Wei et al. [44] demonstrated that both the community structure and temporal succession patterns of culturable PSB during composting were significantly correlated with key physicochemical parameters, including pH, temperature, organic matter content, and dissolved organic carbon/nitrogen ratios. Notably, the phosphorus-solubilizing capacity of laboratory-isolated PSB strains frequently shows limited field performance due to inadequate adaptability under actual environmental conditions. Nevertheless, through strain optimization, microbial consortia formulation, and carrier material immobilization techniques, PSB have emerged as one of the most important functional microorganisms in sustainable agriculture [13,45].

PSMs demonstrate substantial potential in enhancing plant growth through multiple mechanisms. On the one hand, PSMs enhance the availability of soil phosphorus, ensuring sufficient uptake of this essential macronutrient by plants. They achieve this mainly through organic acid secretion (e.g., lactic, citric, and gluconic acids) and proton extrusion, which lowers rhizospheric pH and chelates metal ions (Ca^2+^, Fe^2+^, and Al^3+^), thereby solubilizing mineral-bound phosphates like Ca_3_(PO_4_)_2_ and FePO_4_ [12,13]. Central to this process are genes encoding enzymes responsible for organic acid biosynthesis. For instance, the *gcd* gene, encoding glucose dehydrogenase, drives the oxidation of glucose to gluconic acid, a potent chelator of metal ions [46]. This reaction is cofactor-dependent, requiring pyrroloquinoline quinone synthesized by the *pqq* gene cluster (*pqqE* and *pqqC*) [47]. Proton extrusion, regulated by H+-ATPase genes like *phoU*, further acidifies the microenvironment, directly solubilizing mineral phosphates [48]. These processes are tightly regulated under phosphate-limiting conditions. Low extracellular phosphate levels can activate phosphate starvation response pathways, upregulating *gcd* and *pqq* expression to maximize organic acid production [49]. On the other hand, PSMs can improve soil P availability by enhancing organic P mineralization. This process entails the enzymatic hydrolysis of phosphoester (C-O-P) or phosphoanhydride (C-P) bonds in compounds such as phytate, phospholipids, or organophosphonates [14,15]. Key enzymes and genes include alkaline phosphatases (encoded by *phoA*, *phoD*, and *phoX*), acid phosphatases (encoded by *phoC* and *acpA*), phytases (encoded by *phyA*), and C-P lyases (encoded by *phnJ*), which hydrolyze or cleave organic P compounds in different soil pH conditions [8,50,51]. In addition, the P solubilization processes of PSMs are also regulated through quorum sensing (QS) systems that coordinate gene expression and metabolic cooperation [52]. QS signaling molecules (e.g., AHLs and AIPs) activate the *PhoR*/*PhoB* two-component system under low P condition, upregulating organic acid and phosphatase secretion genes (e.g., *pqq* and *gcd*), while coordinating biofilm formation for enhanced rhizosphere colonization [53,54]. The relevant mechanisms can be used for the optimization of PSMs through synthetic biology approaches, thereby better serving agricultural production.

As two major types of PSMs, PSB and PSF both have the above-stated mechanisms to promote inorganic phosphate solubilization and organic phosphate mineralization. However, they exhibit distinct physiological and molecular characteristics. For instance, PSB primarily rely on proton-driven organic acid secretion, whereas PSF utilizes extensive mycelial networks to facilitate broader acid diffusion [55,56]. Additionally, PSF can extend phosphorus solubilization efficiency through hyphal expansion and direct nutrient translocation to plant roots. Comparative studies indicate that PSF generally demonstrate superior phosphate solubilization capacity compared to PSB and exhibit greater resilience in maintaining phosphorus bioavailability under abiotic stress conditions, such as high salinity or drought [57,58].

Beyond their phosphorus-solubilizing capacity, PSMs often exhibit multiple plant growth-promoting (PGP) traits. A prominent example is the production of indole-3-acetic acid (IAA), a well-known plant hormone that supports root development and overall plant health [59,60]. The secretion of IAA (indole-3-acetic acid) by PSMs primarily involves tryptophan-dependent and tryptophan-independent pathways [61,62,63]. In the tryptophan-dependent pathway, PSMs utilize tryptophan as a precursor and convert it into IAA through the action of tryptophan-inducible enzymes, or further synthesize IAA via tryptophan metabolic intermediates [64]. In the tryptophan-independent pathway, PSMs synthesize indole through their own metabolic pathways and then combine indole with acetic acid or acetyl-CoA to generate IAA [65]. It is worth noting that the synthesis of IAA by microorganisms is also influenced by various factors, including genetic regulatory mechanisms (such as the *ipdC* and *dhaS* genes), plant root exudates (such as phenolic compounds and carbon sources), and environmental factors (such as low-phosphorus stress and acidic pH conditions) [66,67,68]. Conversely, plant root systems provide a favorable environment for PSMs, supplying essential nutrients such as carbon and nitrogen, along with abundant attachment sites. Root exudates also boost biofilm formation in phosphate, solubilizing bacteria, and influence QS signal production, which enhances bacterial colonization in the rhizosphere and P solubilization efficiency [69]. Evidence has shown that PSMs are more enriched in the rhizosphere than in bulk soil [70,71].

The compounds secreted by PSMs in response to low-P conditions also play important roles in enhancing resistance to various environmental stresses. PSMs exhibit diverse functions across different ecological environments and can improve crop tolerance to abiotic stresses such as salinity, heavy metal contamination, and drought. These aspects will be further discussed in the following sections.

## 3. Applications of PSMs in Heavy Metal Remediation and Stress Alleviation

Heavy metal contamination, frequently associated with industrialization and urbanization, has emerged as a critical environmental challenge impeding sustainable development [72]. Heavy metals cannot naturally degrade in soils, and they can accumulate in plants and animals. This makes them a serious threat to ecosystems and human health. [18]. Bioremediation technologies have gained increasing attention due to their eco-friendliness and cost-efficiency [7,73]. Recent advances have revealed that PSMs not only enhance P bioavailability but also employ multifaceted mechanisms to mitigate heavy metal stress in plants, thereby offering innovative solutions for contaminated soil restoration [74,75,76].

### 3.1. Efficacy of PSMs in Heavy Metal Remediation

#### 3.1.1. Application of PSM Inoculants

Conventional P fertilizers (e.g., superphosphate) derived from P ores often contain heavy metal impurities such as Cd and Pb [77]. For instance, Cd concentrations in certain P ores can range from 5 to 100 mg/kg, and these contaminants are not entirely removed during fertilizer production [78]. Long-term application of such fertilizers leads to heavy metal accumulation in soils, posing risks to human health via crop uptake [79]. In contrast, PSM-based biofertilizers offer an environmentally sustainable alternative by reducing heavy metal inputs at the source [80].

The toxicity of heavy metals to microorganisms complicates the application of functional PSMs in contaminated soils. However, significant progress has been made by isolating metal-tolerant PSMs from polluted sites. These strains are capable of colonizing metal-contaminated soils, secreting organic acids (e.g., citric, oxalic, and succinic acids) and extracellular polymeric substances (EPSs), solubilizing P, and immobilizing heavy metals via phosphate precipitation and pH-mediated fixation [81,82]. For example, Yuan et al. [83] isolated several PSB strains from a heavily contaminated site in Guizhou, China, with maximum tolerable concentrations of 500 mg/kg for Pb^2+^ and 400 mg/kg for Cd^2+^. When applied to contaminated soil with calcium phosphate addition, this consortium increased available P by 40.88% and immobilized 80.76% of Pb and 30.81% of Cd. Similarly, Park et al. [84] demonstrated that inoculated *Enterobacter cloacae* dissolved 17.5% of P from phosphate rock in Pb-contaminated soil, achieving a 32.0% Pb immobilization efficiency through pyromorphite formation. Han et al. [85] showed that the inoculation of a PSB (*Klebsiella* sp.) screened from a contaminated site reduced exchangeable Cd by 43.3% in soil and decreased Cd accumulation in wheat grains by 66.7%. These studies highlight the potential of metal-tolerant PSMs as a promising bioremediation tool for sustainable agriculture.

#### 3.1.2. Synergistic Application of PSMs with Other Amendments

Researchers not only screen metal-tolerant strains but also combine PSMs with materials like biochar. This approach boosts microbial colonization and efficacy. Biochar provides a favorable micro-environment that supports microbial survival and activity, thereby improving metal immobilization efficiency [86]. Evaluated the co-application of pig manure-derived biochar and PSMs under mixed Pb (1000 mg/L) and Cd (500 mg/L) stress. The removal efficiencies of Pb^2+^ and Cd^2+^ were 148.77% and 72.27% higher than those achieved by PSMs alone. The biochar also enhanced acid secretion and extracellular electron transfer by PSMs, significantly improving their resistance and remediation potential. Similarly, Zhou et al. [87] demonstrated that the co-application of biochar and PSMs increased Pb removal efficiency to 71.30% and promoted the formation of stable mineral phases (e.g., Pb_5_(PO_4_)_3_OH and Pb_5_(PO_4_)_3_Cl), thereby reducing Pb mobility and bioavailability. Li et al. [88] demonstrated that biochar served as an effective carrier for *Bacillus subtilis*, increasing the survival of the PSB in Cd-contaminated soil, elevating phosphatase activity and altering soil metabolomics. Applying the PSB-loaded biochar increased wheat shoot biomass by 30.3%, P accumulation by 50.4%, and reduced Cd accumulation by 24.1%.

Besides biochar, PSMs have also been co-applied with P-containing materials to enhance remediation efficacy. For instance, Teng et al. [89] synthesized a phosphate-functionalized nanomaterial with a phosphate- and carbon-rich core–shell structure, demonstrating efficient Pb^2+^ adsorption and stable lead phosphate mineral formation. Combining it with PSB (*Leclercia adecarboxylata*) works better for Pb immobilization than using individual nanomaterial or PSB treatment. It increases the residual Pb fraction by 93.94% compared to the control. Qu et al. [90] developed a bio-composite consisting of bone char-supported, carboxymethyl cellulose-stabilized iron sulfide loaded with PSMs (*Enterobacter* sp.) for Pb passivation. This composite achieved a Pb immobilization efficiency of 65.47% via chemical precipitation, complexation, electrostatic attraction, and biomineralization, which was significantly higher than that achieved by PSMs alone (only 0.24%). Li et al. [91] reported that the co-application of iron-doped hydroxyapatite and PSMs (*Ochrobactrum anthropic*) enhanced the formation of stable mineral precipitates through organic acid-mediated release of iron and phosphate, increasing the residual fractions of Cd, Pb, and As by 109.09%, 49.21%, and 25.00%, respectively. This approach improved soil fertility and presented a promising eco-friendly remediation strategy.

#### 3.1.3. Synergistic Effects of PSMs on Plant-Assisted Remediation

Combining PSMs with phytoremediation is another promising approach in the remediation of heavy metal-contaminated soils [92,93]. Phytoremediation is an environmentally sound method that has received considerable attention. However, hyperaccumulator plants often exhibit low biomass and slow growth rates. High concentrations of heavy metals can disrupt plant physiological processes, weaken defense mechanisms, and increase susceptibility to pathogens, ultimately reducing remediation efficiency [77,94]. Plant growth-promoting microorganisms (PGPMs), such as PSMs, can stimulate plant growth and enhance heavy metal tolerance, thereby strengthening the effectiveness of phytoremediation [95,96]. For example, He et al. [97] isolated a PSB strain (*Rahnella* sp. JN6) with high tolerance to Cd, Pb, and Zn, and found that its inoculation significantly enhanced the growth and metal uptake of *Brassica napus*, increasing both biomass and heavy metal accumulation. He et al. [98] studied how inoculating *Acinetobacter pittii* and *Escherichia coli* affects Cd accumulation in *Solanum nigrum* L. They found that it increased Cd uptake by 119% and 88%, respectively, which was due to rhizobacterial community modulation and enhanced Cd mobilization in the rhizosphere.

In summary, PSMs demonstrate multifaceted benefits in heavy metal-contaminated soil remediation and plant stress mitigation. The underlying mechanisms involved will be systematically elucidated in the next section.

### 3.2. Mechanisms of Heavy Metal Remediation in Soil by PSMs

#### 3.2.1. Influence of PSMs on the Physicochemical Status of Heavy Metals in Soil

For plants subjected to heavy metal stress, the primary mechanism of PSMs lies in their capacity to modulate the chemical speciation and bioavailability of heavy metals in soil. The interactions between PSMs and heavy metals in soil are complex and depend on the type of heavy metal, the specific PSMs strains involved, and environmental factors. Nonetheless, the mechanisms can be categorized into three primary pathways (Figure 2):

(1)Heavy Metal Precipitation and Immobilization

The remediation effect of PSMs on heavy metals is closely linked to their phosphate-solubilizing capacity. PSMs can release free phosphate ions into the soil solution, which react with heavy metal ions to form insoluble phosphate precipitates, thereby reducing their bioavailability. For example, *Enterobacter cloacae* can solubilize phosphate. This action promotes the formation of lead–phosphate complexes, thus significantly reducing lead’s mobility and bioavailability [75]. In addition, organic acids (e.g., citric acid and oxalic acid) and EPS secreted by PSMs can chelate and immobilize heavy metals. The functional groups on microbial cell walls, such as carboxyl groups, also play a key role in binding metal ions through surface complexation [83]. For instance, Qin et al. [99] studied the effects of two PSM strains (*Klebsiella* sp. M2 and *Kluyvera* sp. M8) on phosphate uptake efficiency and the suppression of Cd and Pb uptake in radish. The results showed that the strains dissolved inorganic P via secretion of organic acids, and immobilized Cd and Pb through cell wall adsorption and induced precipitation, thereby significantly reducing their accumulation in radish tissues.

(2)Heavy Metal Solubilization and Mobilization

The effects of PSB on heavy metals are influenced by soil properties such as pH, redox condition, and P content. While secreting organic acids to lower soil pH and dissolve insoluble P, PSB may concurrently alter the solubility and bioavailability of heavy metals. In P-deficient soils without exogenous P amendments, PSB application can dissolve heavy metal minerals (e.g., Galena and Otavite), increasing their mobility. This mechanism has been utilized in microbial-assisted phytoremediation to enhance the efficiency of phytoextraction in contaminated soils. For instance, Li et al. [100] isolated a multi-metal-resistant PSB strain from contaminated lake sediments, which significantly enhanced Cu solubilization in polluted soils, increasing Cu uptake in maize and sunflower by 2.8 and 1.7 times, respectively, while also promoting plant growth. He et al. [36] found that in the rhizosphere of *Solanum nigrum* L., PSB inoculation led to higher concentrations of organic acids like malic acid and alanine, which correlated with increased Cd mobilization. Another study by Cheng et al. [101] introduced *Leclercia adecarboxylata* into the rhizosphere of *Potamogeton crispus* L. and observed both reduced soil residual Cd fractions and greater plant Cd accumulation. This showed that introducing specific bacteria can enhance Cd phytoremediation effectiveness.

(3)Redox Transformation of Heavy Metals

Some microorganisms can alter the oxidation states of heavy metals through metabolic processes, thereby reducing their toxicity. For instance, *Cellulosimicrobium cellulans* is capable of reducing highly toxic hexavalent chromium [Cr(VI)] to the less toxic trivalent form [Cr(III)] [102]. *Desulfosporosinus* sp. can oxidize the more toxic arsenite [As(III)] to the less toxic arsenate [As(V)] [103]. Such redox transformations can play a crucial role in assisting plants grown in heavy metal-stressed soils [104]. For instance, Wani et al. [105] reported that the inoculation of a *Bacillus* sp. strain with both phosphate-solubilizing and Cr-reducing capabilities significantly improved the growth, nodulation, chlorophyll content, seed yield, and grain protein content of chickpeas grown in Cr-contaminated soils.

#### 3.2.2. Plant Growth-Promoting Effects of PSMs on Phytoremediation

Phosphorus, as an essential element for plant growth, serves as the material basis for plant stress resistance and defense mechanisms against pests and diseases [106]. Therefore, improving plant P uptake through PSMs application helps to ensure the health of plants used for phytoremediation of heavy metals. Furthermore, plant growth-promoting substances secreted by PSM, such as IAA, ACC deaminase, and siderophores, can help protect plant health under heavy metal stress. For example, Kumar et al. [107] found that PSB (*Enterobacter* sp.) and its siderophore-overproducing mutant, isolated from fly ash-contaminated soil, could significantly enhance the biomass and metal uptake (Ni, Zn, and Cr) of *Brassica juncea* L. grown in fly ash-amended soil by solubilizing phosphate, secreting IAA, and producing siderophores that can chelate heavy metals. Han et al. [85] reported that the application of PSB (*Klebsiella* sp. M2) not only immobilized soil cadmium but also promoted wheat root development by secreting IAA (4.1 µg/mL). Some PSB can also reduce ethylene levels in plants through ACC deaminase activity, alleviating the inhibitory effects of heavy metal stress on plant growth [73,108].

Furthermore, PSMs can boost plant tolerance by increasing the activity of antioxidant enzymes like superoxide dismutase (SOD), peroxidase (POD), and catalase (CAT). These enzymes remove harmful reactive oxygen species (ROS), such as hydroxyl radicals (·OH) and hydrogen peroxide (H_2_O_2_), thereby reducing the oxidative damage to plant cells caused by heavy metals. For instance, Cheng et al. [101] found that after inoculating PSB (*Leclercia adecarboxylata*), the antioxidant enzyme activities in *Myriophyllum aquaticum* significantly increased, with CAT activity rising by 127% and malondialdehyde content decreasing by 64.19%, effectively mitigating lipid peroxidation and improving Cd tolerance.

#### 3.2.3. Indirect Effects of PSMs Through Microbial Community Regulation

Soil microbial communities are shaped by intricate interactions such as competition and cooperation among various microorganisms. The inoculation of PSMs, whether indigenous or exogenous, inevitably disturbs existing microbial ecological networks [109]. Studies demonstrate that PSMs can positively influence the rhizosphere by restructuring microbial community composition and enriching functional microbial taxa, thereby enhancing plant tolerance to heavy metal stress. For example, He et al. [98] reported that the inoculation of PSB strains modulated rhizosphere microbial structure and metabolic functions, enriching microbial taxa associated with Cd mobilization and plant-growth promotion, thereby enhancing Cd phytoextraction by *Solanum nigrum* L. Similarly, Han et al. [85] found that inoculation with PSB (*Klebsiella* sp. M2) reshaped the wheat rhizosphere microbial community by increasing the relative abundance of key bacteria such as *Ramlibacter* and *Microvirga*, which are involved in Cd immobilization and plant-growth promotion, resulting in reduced Cd uptake by plants. He et al. [110] validated this mechanism using fluorescent labeling and synthetic microbial community approaches. Their study revealed that PSB (*Escherichia coli*) colonized both rhizosphere soil and roots of *Suaeda salsa* in saline soils, enriching key functional bacteria and significantly improving Cd accumulation by up to 2.4-fold compared to controls.

## 4. Applications of PSMs in Drought Stress Mitigation

Climate change-induced drought poses severe threats to global agricultural productivity and food security. Drought stress disrupts critical plant physiological functions, including photosynthesis, stomatal regulation, and transpiration [111,112]. As water constitutes 80–95% of plant biomass, prolonged moisture deficits induce stomatal closure, reduce CO_2_ assimilation, and suppress photosynthetic efficiency, ultimately leading to stunted growth and diminished crop yields [113,114].

While soil microorganisms are also vulnerable to drought, certain PSMs, such as *Phyllobacterium* sp., exhibit inherent drought tolerance while retaining their P-mobilizing capacity [115]. In addition to facilitating the dissolution of insoluble soil P, PSMs can enhance plant growth and survival under drought conditions through multiple strategies, such as promoting root development, boosting antioxidant capacity, and improving soil aggregation [116]. Consequently, the application of PSMs to support plant drought tolerance has attracted increasing attention. This section reviews the current research progress on the application of PSMs under drought stress, focusing on their effectiveness and underlying mechanisms.

### 4.1. Promotion of Root Growth and Nutrient Uptake

Phosphorus is a critical nutrient for plant growth and stress resilience, as it plays essential roles in photosynthesis, respiration, and antioxidant defense systems [117]. However, soil particles like iron or aluminum oxides, and calcium salts often bind phosphorus. This makes it hard for roots to absorb phosphorus through passive diffusion [118]. Consequently, plant P uptake relies primarily on active root absorption, placing plants at a heightened risk of P deficiency under drought conditions [119].

PSMs play a dual role in addressing this issue. Firstly, PSMs solubilize fixed P in soils, increasing P availability for plants under drought stress [120,121]. Secondly, PSMs secrete plant growth-promoting substances such as IAA, which stimulates root growth and development, enabling plants to absorb water and nutrients more effectively [122]. Research indicates that IAA also promotes organic acid synthesis, enhancing PSMs’ phosphate-solubilizing ability under stress [123]. Empirical studies have demonstrated the efficacy of PSMs in facilitating root development for drought resilience. For instance, Yahya et al. [124] reported that the inoculation of a PSM consortium increased root length, surface area, and tip density in wheat by 24%, 29%, and 46%, respectively. The improvements in root architecture and exudate composition enhanced overall nutrient uptake efficiency under drought conditions. Similarly, Zolfaghari et al. [125] found that inoculation with indigenous PSMs (*Microbacterium* sp. and *Streptomyces* sp.) significantly increased root length, root weight, and relative water content in *Quercus brantii* seedlings subjected to drought in a Mediterranean climate.

In addition, the mutualistic relationship between arbuscular mycorrhizal fungi (AMF) and PSMs can further improve plant P acquisition under drought since AMF effectively expand the root absorption area [126,127]. For example, Ghorchiani et al. [128] found that co-inoculation with PSMs (*Pseudomonas fluorescens*) and AMF (*Funneliformis mosseae*) under drought conditions significantly increased the levels of P and nitrogen, and chlorophyll content, thereby alleviating drought stress and improving yield. Similarly, Nacoon et al. [129] examined the effects of PSB (*Burkholderia vietnamiensis*) and AMF (*Rhizophagus aggregatus*) on *Helianthus tuberosus* under drought conditions, finding that the combined treatment enhanced P uptake and improved plant physiological traits such as root length, chlorophyll content, and soluble sugar content. These studies demonstrate that AMF-PSMs partnerships not only enhance phosphorus acquisition but also improve key physiological traits, ultimately bolstering plant drought tolerance and productivity [130,131].

### 4.2. Enhancement of Plant Antioxidant Capacity and Osmotic Regulation

Drought stress triggers the accumulation of ROS in plants, inducing oxidative damage to cellular membranes and biomacromolecules [132]. Research demonstrates that PSMs can enhance plant antioxidant defense mechanisms through multiple pathways, helping to mitigate the adverse effects of drought stress. Firstly, PSB indirectly regulate antioxidant enzyme activities by improving plant nutrition: P—as a critical component of nucleic acids, ATP, and membrane phospholipids—ensures optimized energy metabolism and membrane stability when adequately supplied, thereby alleviating drought-induced oxidative stress [13]. PSMs also enhance micronutrient acquisition (e.g., iron via siderophores secretion), where iron serves as a cofactor for SOD that accelerates ROS scavenging [133]. Secondly, PSB-derived metabolites like oxalic and lactic acids function as signaling molecules, activating plant defense responses through calcium signaling pathways that balance ROS production and elimination [134]. Additionally, phytohormones secreted by PSMs (e.g., IAA and gibberellins) can modulate the biosynthesis of antioxidant enzyme (including CAT and ascorbate peroxidase (APX)) via gene expression modulation, thereby reinforcing stress tolerance [135,136].

These synergistic mechanisms collectively enhance plant resilience to drought-induced oxidative stress [137]. For instance, Sahandi et al. [138] demonstrated that inoculating peppermint (*Mentha piperita* L.) with PSB (*Pseudomonas putida* and *Pantoea agglomerans*) increased antioxidant enzyme activities (CAT, POD, APX, and polyphenol oxidase (PPO)) by 20–50% and significantly improved the growth and phytochemical characteristics of peppermint. Similarly, Kang et al. [139] reported that PSB (*Enterobacter ludwigii* AFFR02 and *Bacillus megaterium* Mj1212) inoculation in alfalfa under drought condition elevated total phenolic content by 46.08%, DPPH-scavenging activity by 39.66%, and SOD activity by 28.51%, while reducing abscisic acid levels by 24.41%. Azizi et al. [140] also reported that a biofertilizer combining *Pseudomonas putida*, *Pantoea agglomerans*, and seaweed extract elevated relative water content by 40% and chlorophyll biosynthesis by 46% in drought-stressed *Calendula officinalis*, while enhancing antioxidant enzyme activities (e.g., CAT by 127% and APX by 38%), thereby significantly mitigating the adverse effects of drought stress.

Furthermore, the accumulation of osmolytes such as proline, soluble sugars, and betaine is a key mechanism by which plants cope with drought stress [141]. PSMs enhance the P supply and improve photosynthetic efficiency, thereby promoting the accumulation of these photosynthetic products. For instance, Chandra et al. [142] studied how PSB (*Pseudomonas* spp.) affects finger millet (*Eleusine coracana*) under drought stress. They found that PSB inoculation increased plant proline by 1.56 times and lowered malondialdehyde (MDA, a marker of lipid peroxidation) by 53.5%. It also boosted ACC deaminase activity, which helps reduce stress-induced ethylene. These changes help plants better tolerate drought conditions. Similarly, Hu et al. [143] reported that the PSB consortium inoculation in *Parashorea chinensis* seedlings considerably enhanced antioxidant enzyme activity (SOD, CAT, and POD) by 9.83–292.32%. It also increased proline, soluble sugars, and soluble protein by 61.41%, 43.86%, and 45.48% respectively. As a result, this significantly improved the plants’ osmotic regulation and overall stress tolerance.

### 4.3. Formation of Biofilm and Soil Aggregation

PSMs, particularly filamentous fungi, can form biofilms rich in polysaccharides [144,145,146]. Under drought stress, certain microorganisms increase EPS production, enhancing their adhesion and stress resistance while improving soil physical structure, aggregation, and water retention. For example, Thakuria et al. [147] demonstrated that inoculating soil with *Bacillus megaterium* increased the proportion of macroaggregates (>250 μm) while reducing microaggregates (<53 μm), thereby stabilizing soil structure and enhancing water holding capacity. Similarly, Beheshti et al. [148] observed that biofilm-forming PSMs in rice rhizospheres facilitated soil aggregation and activated occluded P in periphytic biofilms, supporting rice growth during drought.

Plant–microbe interactions further amplify these benefits. Lucero et al. [149] reported that root exudates from peanuts, maize, and soybeans stimulated biofilm formation in phosphate-solubilizing endophytes (*Serratia* sp. S119 and *Enterobacter* sp. J49), enhancing plant stress tolerance and growth promotion. This mutualistic relationship underscores the role of biofilms in facilitating nutrient exchange and improving soil–plant water dynamics under drought conditions.

## 5. Applications of PSMs in Saline–Alkaline Stress Mitigation

Soil salinization represents one of the most significant challenges confronting global agricultural production, posing a severe threat to food security and agricultural sustainability [150,151]. Saline–alkaline soils exhibit poor soil structure and fertility, which inhibit plant growth and significantly reduce crop production [152]. To maintain acceptable yields, agriculture in saline regions often requires excessive fertilizer application, consequently leading to increased production costs and non-point-source pollution from nitrogen and P runoff [153].

Salt stress adversely affects plant growth and metabolism through multiple mechanisms, such as ion imbalance, osmotic stress, and oxidative stress [154,155,156]. Notably, PSMs can effectively mitigate these negative effects through various physiological and biochemical mechanisms. Consequently, numerous researchers have focused on screening salt-tolerant PSMs from saline environments and developed them into functional inoculants (Table 1), which have emerged as an important biological tool for the improvement and utilization of saline–alkaline soils.

### 5.1. Enhancement of Microbial-Mediated Soil Nutrient Cycling

Microorganisms play a crucial role in the biogeochemical cycles in soils, driving essential processes such as nutrient cycling, organic matter decomposition, and soil health maintenance [164,165]. However, the activity of soil microorganisms is suppressed under saline–alkaline stress, which is a key reason for the low fertility of saline–alkaline soils and their inability to effectively supply plant nutrients [166]. The isolation and application of halo-tolerant PSMs as biofertilizers can significantly enhance microbial-mediated biochemical processes [167].

The promotion of phosphate solubilization in soil by PSMs is particularly crucial in saline–alkaline conditions. In soils with high pH and abundant calcium ions, phosphates tend to form stable Ca-P minerals, and applied P fertilizers are quickly fixed, resulting in low utilization efficiency [168]. PSMs can lower the micro-environmental pH through mechanisms such as organic acid secretion, dissolve insoluble P minerals, and increase the amount of plant-available P [169,170]. For example, Saranya et al. [14] reported that *Curtobacterium luteum* can increase the availability of P in saline–alkaline soils through the secretion of organic acids (such as tartaric and malic acids), phosphatases, and phytases. The inoculation of this strain remarkably increased the population of PSB in soils, reduced the soil pH from 7.69 to 6.90, and increased the P content in rice-growing soils from 105.41 mg/kg to 176.28 mg/kg. Adnan et al. [171] also reported that the inoculation of a PSB consortium in calcareous soils counteracted the antagonistic effects of lime on P availability by enhanced soil acidification and significantly improved wheat growth.

Beyond P solubilization, the addition of PSMs also promotes other nutrient cycling processes in saline–alkaline soils. For instance, Li et al. [172] demonstrated that the inoculation of halo-tolerant PSB (*Bacillus paramycoides*) not only increased soil P availability by 12.5%, but also boosted soil urease activity and available nitrogen content, thereby promoting the growth of wheat seedlings in saline–alkaline soil. Moreover, saline–alkaline soils often suffer from high bulk density, poor porosity, and sodium-induced dispersion, which reduce water retention and aeration [173]. PSMs, especially those with high EPS production, can improve soil structure and enhance the aeration and water retention performance of saline–alkaline soils through the secretion of organic substances that promotes soil aggregation [119,174].

### 5.2. Plant-Growth Promotion and Antioxidant Effects of PSMs Under Salt Stress

Beyond improving soil quality, the ability of PSMs to regulate plant growth and antioxidant systems is crucial for mitigating salt stress in plants. Researchers have isolated salt-tolerant PSM strains from saline environments that produce phytohormones such as IAA, which promote cell elongation and division, mitigating the growth-inhibitory effects of salt stress [14,175,176]. For example, Mahdi et al. [157] isolated a salt-tolerant PSB, *Enterobacter asburiae* QF11, from the rhizosphere of quinoa. This bacterium could produce up to 795.31 µg/mL of IAA and significantly enhanced root elongation and seedling development in quinoa after inoculation.

Salt stress causes physiological and biochemical changes in plants, leading to excessive ROS accumulation. These changes include stomatal closure, impaired photosynthesis, activated plasma membrane NADPH oxidases, and increased electron leakage in the photosynthetic electron transport chain. The buildup of ROS induces lipid peroxidation in membranes and damages key enzymes and nucleic acids [177]. PSMs can mitigate these oxidative stresses by bolstering plants’ antioxidant defense mechanisms, as discussed in Section 3.2, thereby alleviating salinity-induced oxidative stress. For instance, Joe et al. [178] investigated two salt-tolerant endophytic PSMs, *Acinetobacter* sp. ACMS25 and *Bacillus* sp. PVMX4, isolated from the root of *Phyllanthus amarus*. These bacteria significantly improved the plant’s antioxidant capacity under salt stress by increasing phenolic content (by 21%), free radical-scavenging ability (DPPH and ABTS assays, by 15–19%), and antioxidant enzyme activities (SOD, APX, and PPO, by 19–31%).

With the intensification of climate change and water scarcity, the problem of soil salinization is likely to deteriorate further worldwide [179,180]. Particularly in arid and semi-arid regions, agricultural production may face the dual challenges of drought and salinity. Similarly, soils contaminated with heavy metals in saline–alkali and arid regions also present cross-stressor synergies. Different abiotic stressors often exhibit overlapping detrimental effects on plants, such as ROS accumulation due to cellular damage and impaired nutrient uptake resulting from root system dysfunction [177,181]. The stress-alleviation mechanisms of PSMs may play a particularly significant role under such conditions [181,182,183]. For instance, the production of EPS can simultaneously chelate heavy metals and enhance drought resistance, enabling PSMs to survive and function effectively under complex stress conditions [108,119]. Osman et al. [141] reported that inoculating PSB (*Bacillus megatherium* var. phosphaticum) alleviated osmotic stress in sugar beet caused by drought and soil salinity, thereby enhancing its growth and quality under moderate drought in salt-affected soils.

### 5.3. Regulation of Plant Ion Homeostasis

Under saline stress conditions, elevated concentrations of Na^+^ and Cl^−^ ions in soil solution significantly impair plant-root water-uptake capacity [184]. The concomitant increase in Na^+^ concentration competitively inhibits K^+^ absorption, a critical cation involved in multiple physiological processes, including enzyme activation, osmotic regulation, and membrane potential maintenance [185]. This ionic imbalance typically manifests as decreased K^+^/Na^+^ ratios, subsequently disrupting cellular homeostasis [154,186].

Recent investigations reveal that salt-tolerant PSMs employ multifaceted mechanisms to mitigate salinity stress. Through the dissolution of calcium phosphate minerals, PSMs release Ca^2+^, while their EPS production reduces Na⁺ uptake, thereby improving K^+^/Na^+^ ratios [108,187]. Certain PSMs demonstrate the capacity to upregulate key ion transporter genes, including HKT1 for K^+^ homeostasis, NHX7 for Na^+^ compartmentalization, and H^+^-PPase for Na^+^ sequestration [188,189,190]. For instance, Safdarian et al. [188] reported that *Enterobacter asburiae*, isolated from the rhizosphere of halophytes, upregulated the expression of H^+^-PPase, HKT1, NHX7, CAT, and APX genes in the root of inoculated salt-stressed plants, reducing leaf Na^+^/K^+^ ratios by 41–58% and enhancing plant antioxidant defenses. Similarly, *Bacillus licheniformis* QA1, isolated from the rhizosphere of quinoa, demonstrated remarkable ion balancing capacity by reducing Na^+^ content by 63.63% while increasing K^+^ by 32.55% in treated plants [157].

These findings establish PSMs as biotechnological tools for salinity stress mitigation, offering sustainable alternatives to conventional soil amendment methods. The synergistic effects of ionic modulation, genetic regulation, and growth promotion highlight the potential of PSMs to address the complex challenges posed by soil salinity in agricultural systems.

## 6. Challenges and Future Perspectives

PSMs hold significant promise for enhancing agricultural sustainability through improved plant stress tolerance. However, their widespread application faces critical challenges that require interdisciplinary solutions.

(1)Elucidation of Underlying Genetic Mechanisms

A fundamental limitation lies in the incomplete understanding of molecular mechanisms governing PSM–plant interactions. While PSMs are known to modulate antioxidant enzyme activity, the precise regulatory pathways—such as metabolite interactions with plant gene promoters or stress-responsive genetic adaptations in PSMs—remain poorly characterized. Advanced methodologies integrating multi-omics (metagenomics and metabolomics) and gene-editing tools (e.g., CRISPR-Cas9) are essential to map metabolic networks and identify key functional genes driving these processes [191,192].

(2)Targeted Screening and Optimization of PSM Strains

Equally critical is the development of robust strategies for strain selection under complex environmental conditions. Conventional screening methods, often limited to single stressors like salinity or heavy metals, fail to address real-world scenarios involving combined abiotic stresses (e.g., heavy metal pollution in saline–alkaline soils). Emerging technologies such as high-throughput microfluidic platforms, which simulate multifactorial stress environments (pH fluctuations, osmotic pressure, and pollutant gradients), coupled with machine-learning algorithms analyzing genotype–phenotype correlations, could revolutionize strain optimization [193]. Synthetic biology approaches further offer opportunities to engineer PSMs with enhanced organic acid biosynthesis pathways or stress-tolerance traits [110,194,195]. In addition, while PSF are more suitable for long-term stress alleviation owing to their metabolic versatility and functional stability, PSB can better enhance short-term stress resistance by secreting hormones and dissolving phosphorus rapidly; thus, their synergistic application holds significant potential for developing innovative solutions in sustainable agriculture [196,197,198].

(3)Synergism with Other Biological and Agronomic Managements

Environmental variables exert profound influences on PSM efficacy [14]. In alkaline soils, where P forms insoluble calcium phosphate complexes, PSMs demonstrate pH-modulating capacity, yet their performance declines at extreme alkalinity. Similarly, an elevated C/P ratio directs microbial activity toward carbon metabolism, suppressing P solubilization [199]. Single-strain PSM applications often exhibit limited efficacy, whereas microbial consortia (e.g., PSMs + nitrogen-fixing bacteria + cellulose decomposing microorganisms) can significantly improve resilience and overall performance [132,156]. Future studies should further test synergistic strategies combining PSMs with other beneficial microorganisms or soil amendments (e.g., biochar and cover crop), which may enhance microbial survival and overcome environmental disadvantages.

(4)Long-Term Ecological Assessment

The ecological consequences of PSMs’ deployment require comprehensive evaluation through longitudinal studies. Empirical evidence indicates that recurrent PSM inoculation may disrupt native soil microbiomes, as demonstrated by reductions in diversity indices of microbial communities across multiple cropping systems [174,200]. Horizontal gene transfer of functional markers (e.g., phoD) among soil microbiota further underscores the need for ecological risk assessments [200,201]. Long-term (>10-year) field trials incorporating metagenomic monitoring and ecological modeling should quantify impacts on soil biodiversity, nutrient-cycling stability, and crop yield consistency.

(5)Large-Scale Production and Commercialization

Translating PSMs research into agricultural practice requires addressing technical and socioeconomic barriers. Large-scale production faces challenges in maintaining microbial viability during fermentation and storage, demanding innovations in lyophilization and encapsulation technologies [202,203]. Concurrently, farmer adoption hinges on demonstrating economic viability by optimizing application timing, dosage, and compatibility with conventional practices (e.g., irrigation schedules and fertilizer blends). Regulatory frameworks must evolve to standardize biofertilizer certification while ensuring supply chain integrity for perishable microbial products.

## 7. Conclusions

PSMs demonstrate significant potential in mitigating soil adversities, such as heavy metal pollution, drought, and salinity stress (Figure 3). PSMs share certain core mechanisms in mitigating plant stress under various abiotic adversities. By secreting organic acids, PSMs chelate heavy metal ions, reducing their bioavailability. They also solubilize P through phosphatase secretion, converting insoluble P into plant-available forms and enhancing P uptake efficiency. Additionally, PSMs produce plant hormones and enzymes, promoting root growth and improving plant antioxidant defenses. These mechanisms collectively enhance plant growth and yield in soils affected by heavy metals, drought, and salinity.

However, challenges remain in the application of PSMs. Future research should focus on screening and optimizing highly efficient PSMs strains with strong stress tolerance, exploring their interaction mechanisms with plants, and enhancing their performance through environmental factor control. Investigating synergistic applications of PSMs with other beneficial microorganisms is also crucial. Furthermore, field trials assessing PSMs effectiveness across diverse soil types and cropping systems are essential for advancing sustainable agriculture.

In summary, PSMs represent a sustainable and multifaceted approach to mitigating abiotic stress in plants through soil–plant–microbe interactions. Their application prospects are broad, and future research should prioritize strain optimization, mechanism exploration, and synergistic strategies to maximize their value in agricultural production and support sustainable development.

## Figures and Tables

**Figure 1 plants-14-01558-f001:**
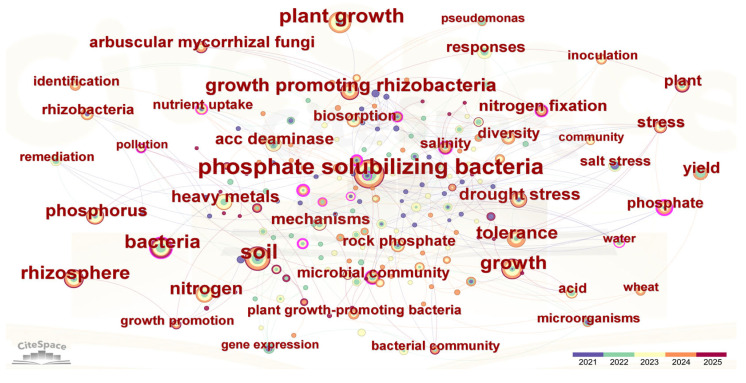
Keyword co-occurrence network of phosphate-solubilizing microorganisms (PSMs) in relation to abiotic stress mitigation, generated using CiteSpace. The analysis was based on 261 articles retrieved from the Web of Science Core Collection (2021–2025) using the following search query: (“phosphate solubilizing” OR “phosphorus solubilizing”) AND (“microorganism” OR “bacteria” OR “fungi” OR “actinomycetes” OR “archaea”) AND (“stress tolerance” OR “salinity” OR “drought” OR “heavy metal” OR “pollution” OR “environmental stress”) AND (“plant” OR “agriculture” OR “soil”). Nodes represent keywords, with node size indicating frequency of occurrence and node color indicating the time span of occurrence; lines represent co-occurrence relationships. CiteSpace settings: time span = 2021–2025; slice length = 1 year; g-index (k = 25); LRF = 2.5; L/N = 10; LBY = 5; e = 1.0. The resulting network includes 210 nodes and 388 links, with a density of 0.0177 and a largest connected component comprising 208 nodes (99%).

**Figure 2 plants-14-01558-f002:**
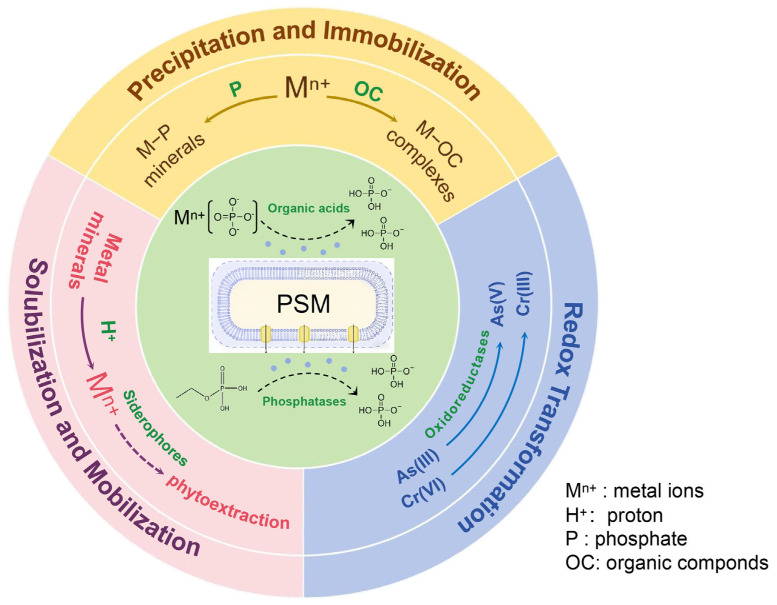
Mechanisms by which phosphate-solubilizing microorganisms (PSMs) influence heavy metal dynamics in soil. Note: This diagram illustrates three primary mechanisms through which PSMs affect heavy metal (M) behavior in soil. (1) Precipitation and immobilization: Phosphate (P) and organic compounds (OCs) released by PSMs react with metal ions (M^n+^) to form stable metal–phosphate (M−P) minerals and metal–organic complexes (M−OCs), reducing metal bioavailability. (2) Solubilization and mobilization: PSMs secrete organic acids and siderophores, and lower pH via proton (H^+^) release, enhancing dissolution of metal minerals. Mobilized metals can be taken up by plants (phytoextraction). (3) Redox transformation: PSMs produce oxidoreductases that catalyze redox transformations of metals such as As(V)/As(III) and Cr(IV)/Cr(VI), altering their mobility and toxicity. Arrows indicate the direction of metal transformation or movement. The inner box depicts a typical PSM cell excreting functional metabolites.

**Figure 3 plants-14-01558-f003:**
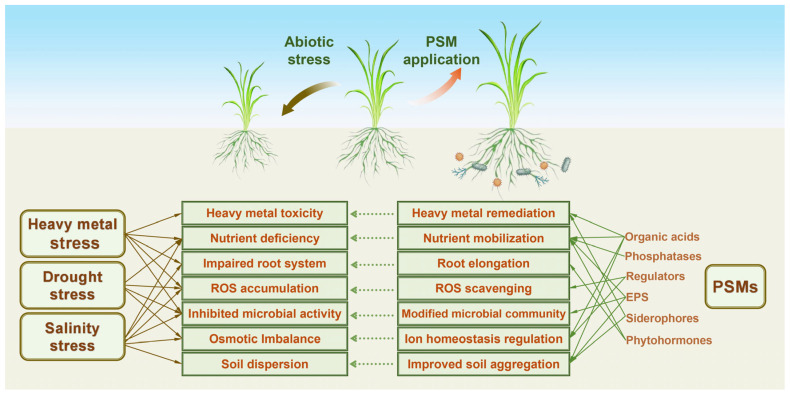
Schematic representation of the correspondence between the mechanisms of abiotic stress-induced damage in plants and the mitigating effects of PSMs. Abiotic stress factors negatively affect plant growth through various direct and indirect mechanisms, such as oxidative stress, ion toxicity, and nutrient imbalance. PSMs alleviate these effects through the secretion of organic acids, phytohormones, EPS, and enzymes that enhance nutrient availability and stress tolerance. Directional arrows illustrate the functional alignment between stress-induced damage pathways and the corresponding microbial countermeasures.

**Table 1 plants-14-01558-t001:** Summary of selected studies on PSM applications under saline–alkaline conditions.

PSM Strain	Strain Source	Plant Species	Effects	Mechanisms	Reference
*Bacillus pumilus*	Quinoa fields in Morocco	*Chenopodium quinoa*	Seed germination rate increased by 305%; seedling length increased by 211%	Organic acid secretion; IAA and siderophores production; biofilm formation	[157]
*Kushneria* sp.	Saline soil on the Coast of Yellow Sea of China	*Suaeda salsa*	Available phosphorus increased by more than 10 times; plant height and biomass increased by 1.5–10 times	Organic acid secretion	[158]
*Arthrobacter* sp.; *Bacillus* sp.	Rhizosphere of maize in Cameroon	*Solanum lycopersicum*	Plant height increased by 24.1%; dry weight increased by 73.5%; total biomass increased by 115%	Organic acid secretion	[159]
*Bacillus megaterium*	Rhizosphere of Tamarix ramosissima in Mexicali valley	*Phaseolus vulgaris*	Root length increased by 151%; root dry weight increased by 188%; phosphorus content increased by 114%	Organic acid secretion; enhanced photosynthesis	[160]
*Bacillus pumilus; Bacillus amyloliquefaciens*	Laboratory collection	*Cicer arientnum*	Shoot dry weight increased by 34%; leaf phosphorus content increased by 600%; total chlorophyll content increased by 32%	Organic acid secretion; phosphatase production; H^+^-ATPase activation; enhanced antioxidant system	[161]
*Pseudomonas azotoformans*	Agricultural field in southern Algeria	*Triticum aestivum*	Wheat seed germination rate increased to 68.88%; fresh weight increased by 99.68%	Organic acid secretion; phosphatase production	[162]
*Bacillus* sp.; *Burkholderia* sp.	Laboratory collection	*Zea mays*	Corn height increased by 5.6%; shoot biomass increased by 7.8%	Organic acid secretion; IAA, siderophore, and axine production	[163]

## Data Availability

No new data were created or analyzed in this study. Data sharing is not applicable to this article.

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
