# Peer review of "Alleviation of Plant Abiotic Stress: Mechanistic Insights into Emerging Applications of Phosphate-Solubilizing Microorganisms in Agriculture"

_plants, 2025, doi:10.3390/plants14101558_

Round 1
Reviewer 1 Report
Comments and Suggestions for Authors
The manuscript is comprehensive, well-structured, and offers a timely review of phosphate-solubilizing microorganisms for abiotic stress mitigation. It synthesizes a large volume of recent literature, includes mechanistic insights, and identifies knowledge gaps, elements crucial for a high-impact review. However, some revisions are needed, particularly to improve figure quality, language clarity, and deepen some mechanistic discussions.
Minor points
- Figure 1 is unreadable. The resolution is too low for print or digital viewing. Many nodes and labels are illegible (especially upper left). It fails to visually support the accompanying text.
- Figure 2 has a vague legend ("Mechanisms of PSM influencing heavy metals in soil") and no explanation of components, symbols, or directional arrows. The reader cannot interpret the pathways without text reference.
- Figure 3 has also a vague legend. In addition, Figure 3 includes an unexplained figure element—a small human icon. This is inappropriate in a scientific schematic unless clearly explained. Its meaning is ambiguous and should either be explained or removed.
- Some mechanistic explanations remain superficial or repetitive. For example, while organic acid secretion is repeatedly cited as a key PSM trait, specific gene-level regulation, signal transduction, or quantitative data are often missing. Including even a simplified regulatory pathway (e.g., IAA synthesis) would add depth.
- Cross-stressor synergies (e.g., salinity + heavy metals) are acknowledged but not deeply discussed. This would elevate the article's novelty.
- Line 14: Change “due to their capacities of converting...” to “due to their capacity to convert…”
- Frequent overuse of passive voice and long sentences impairs readability.
Author Response
The manuscript is comprehensive, well-structured, and offers a timely review of phosphate-solubilizing microorganisms for abiotic stress mitigation. It synthesizes a large volume of recent literature, includes mechanistic insights, and identifies knowledge gaps, elements crucial for a high-impact review. However, some revisions are needed, particularly to improve figure quality, language clarity, and deepen some mechanistic discussions.
Response: Thank you for taking the time to review our manuscript and for your valuable comments. We have carefully revised the manuscript to address your concerns and highlighted the corresponding changes in yellow in the revised manuscript. We hope that you will find the revised version satisfactory.
Minor points
- Figure 1 is unreadable. The resolution is too low for print or digital viewing. Many nodes and labels are illegible (especially upper left). It fails to visually support the accompanying text.
Response: We appreciate your valuable comment regarding the quality and legibility of Figure 1. To address this concern, we have regenerated the figure at a significantly higher resolution to ensure clarity for both print and digital formats. Additionally, we have removed the textual metadata from the upper left of the figure and integrated all key parameters (e.g., timespan, selection criteria, network size, and density) directly into the revised figure legend for clarity and accessibility (Lines 60-70). We hope the improved visualization and expanded legend now meet the standards for readability.
- Figure 2 has a vague legend ("Mechanisms of PSM influencing heavy metals in soil") and no explanation of components, symbols, or directional arrows. The reader cannot interpret the pathways without text reference.
Response:We agree that the original legend lacked sufficient detail to guide the interpretation of Figure 2. We have revised the figure legend to explicitly describe the depicted mechanisms (precipitation/immobilization, solubilization/mobilization, and redox transformation), the meaning of abbreviations (e.g., M, P, OC), and the function of directional arrows and symbols. This clarification will help readers better understand how PSMs influence heavy metal dynamics in soil. The revised legend now accompanies the updated manuscript (Lines 289-300).
- Figure 3 has also a vague legend. In addition, Figure 3 includes an unexplained figure element—a small human icon. This is inappropriate in a scientific schematic unless clearly explained. Its meaning is ambiguous and should either be explained or removed.
Response:Thank you for your valuable comment. We have removed the previously included human icon, which was originally intended to represent the role of PSMs in combating the negative effects of abiotic stressors on plants. To convey this concept more appropriately, we have replaced the icon with directional arrows that clearly illustrate the correspondence between stress-induced damage mechanisms and the microbial countermeasures. Additionally, we have revised the figure legend to provide a more detailed and precise explanation of the components and processes depicted in the schematic (Lines 678-684).
- Some mechanistic explanations remain superficial or repetitive. For example, while organic acid secretion is repeatedly cited as a key PSM trait, specific gene-level regulation, signal transduction, or quantitative data are often missing. Including even a simplified regulatory pathway (e.g., IAA synthesis)would add depth.
Response:Thank you for your valuable comment. we have substantially expanded our discussion of the genetic and molecular regulation underlying key PSM traits. Specifically: (1) Lines 122-150 now provides genetic mechanisms governing organic acid and phosphatase secretion; (2) Lines 161-175 incorporates a simplified but comprehensive IAA biosynthesis pathway. These additions address the gap in gene-level regulation while maintaining appropriate scope for a review article. We believe these enhancements significantly strengthen the manuscript's mechanistic foundation.
- Cross-stressor synergies (e.g., salinity + heavy metals) are acknowledged but not deeply discussed. This would elevate the article's novelty.
Response:Thank you for this valuable suggestion. We have expanded our discussion of this aspect in the revised manuscript (Lines 568-581).
- Line 14: Change “due to their capacities of converting...” to “due to their capacity to convert…”
Response:Revised (Line 14). Thank you.
- Frequent overuse of passive voice and long sentences impairs readability.
Response:Thank you for your constructive feedback. We have carefully revised the manuscript to improve readability by systematically replacing passive voice with active constructions where appropriate and breaking down long sentences into clearer, more concise statements. Examples of these edits are lines 190-191, 224-225, 245-247, 270-274, 305-307, 326-332, 360-364, 406-408, 471-481 and 555-558. We believe these changes have significantly enhanced the manuscript's clarity.
Reviewer 2 Report
Comments and Suggestions for Authors
1. Wang et al. provide valuable insights for alleviation stress effects on plants using phosphate solubilizing microorganisms. Based on the title, the review is related to all type of microorganisms; however, most of the content refers to bacteria. The main question addressed in this review is immense, since all types of microorganisms (and their mechanisms) are under investigation for review purposes, for three - distinctively different - types of abiotic stress (heavy metal contamination, drought and soil salinization). The topic is original but needs an in depth analysis since many reviews with microbial application data are available (https://doi.org/10.1080/00103624.2022.2142238; https://doi.org/10.1007/978-81-322-2286-6_12 ; https://doi.org/10.3390/plants11162119 ). Conclusion of the manuscript are well justified; references are appropriate but not adequate for reviewing the topic.
2. A section with general attributes of PSMs is missing, prior analyzing the PSMs effects per type of plant stress. Some potential content information in points (which is not restrictive) is cited below:
- Promotion of plant growth by phosphate solubilizing fungi.
- Information for culturable and non-culturable phosphate-solubilizing microorganisms, emerging solid routes of large-scale availability and microbial applicability.
- Mechanistic quorum sensing (QS) signaling of phosphate-solubilizing bacteria and its potential benefits for agriculture.
- Microbial activity is not independent of plant presence; root exudates stimulate the phosphorus solubilizing ability (as example, bacteria: https://doi.org/10.1038/s41598-023-30915-2). This is a valuable, sustainable insight in using PSMs in agriculture. In lines 294-296 of the text, it is described the effect of PSM to plants without to be mentioned the reverse-way interaction (e.g. the nutritional support of the plant towards PSMs).
- Molecular mechanisms of sensing, transport, signaling and regulating phosphate solubilization in microorganisms (bacteria and fungi); specialized mechanisms for each stress factor could be described at a later stage.
- Physiological, molecular and biochemical differences between phosphate-solubilizing bacteria and/or fungi and how these attributes can affect positively stress alleviation process are not mentioned.
- Direct and indirect modes of for P-solubilization by the microorganisms (Inorganic and organic P-solubilization). Direct and indirect contribution of microorganisms to plant P-uptake.
- Environmental restrictions of using phosphate solubilizing microorganisms; the manuscript is mostly focused on success stories and positive approaches.
- Phosphate-solubilizing microorganisms from air in agriculture (as example https://doi.org/10.3389/fmicb.2020.585215 ).
3. Citespace analysis did not enclose the term ”plant” and/or “agriculture” for the keyword co-occurrence network. Adding keywords emerges subsections of the review topic which are not discussed. Τhe figure 1 projects literature data analysis from 125 references; this is a very small number of publications for review. There are many research articles (not reviews) with close to 100 references in this journal (https://doi.org/10.3390/plants14091268 with related to the topic abiotic stress). In a broader sense, similar research field reviews e.g. https://doi.org/10.3390/plants14060865 and https://doi.org/10.3390/plants11162119 encloses equal or more than 200 references.
4. Keyword clustering information is missing from the explanation of the figure 1 and/or manuscript text.
5. Line 64 “Emerging keywords like "acc deaminase" and "biosorption" suggest a growing interest in mechanistic pathways and remediation processes”: There is no documentation cited if this information is plant- or microbial based since both type of organisms can produce the specific compound (see for bacteria https://doi.org/10.1016/j.micres.2020.126439 ; for plants table 3 https://doi.org/10.3389/fmicb.2015.00937 ).
Author Response
- Wang et al. provide valuable insights for alleviation stress effects on plants using phosphate solubilizing microorganisms. Based on the title, the review is related to all type of microorganisms; however, most of the content refers to bacteria. The main question addressed in this review is immense, since all types of microorganisms (and their mechanisms) are under investigation for review purposes, for three - distinctively different - types of abiotic stress (heavy metal contamination, drought and soil salinization). The topic is original but needs an in depth analysis since many reviews with microbial application data are available (https://doi.org/10.1080/00103624.2022.2142238; https://doi.org/10.1007/978-81-322-2286-6_12 ; https://doi.org/10.3390/plants11162119 ). Conclusion of the manuscript are well justified; references are appropriate but not adequate for reviewing the topic.
Response:We sincerely appreciate your insightful comments and constructive suggestions. In response to the concerns raised, we have implemented the following key improvements:
(1) Scope and Coverage:​​
While PSB naturally dominate both in environmental abundance and existing literature (Lines 97-105), we have significantly enhanced the discussion of PSF and added comparisons between PSB and PSF in stress response strategies to provide a more comprehensive view (Lines 151-160, 629-634).
(2) Mechanistic Depth:​​
We have expanded genetic-level explanations for core processes such as organic acid and phosphatase secretion pathways (Lines 122-150) and IAA biosynthesis and regulation (Lines 161-175). We also highlighted critical knowledge gaps and call for more in-depth exploration in plant-PSM interactions under abiotic stress in the Future Perspectives section (Lines 610-618).
(3) Literature Support:​​
We have strengthened evidence for key claims through targeted reference additions. We incorporated 83 additional references during the revision (total now 203).
These revisions have substantially improved the manuscript's balance and scientific rigor while maintaining its comprehensive scope. We are grateful for the opportunity to enhance our work and hope these modifications address your concerns satisfactorily.
- A section with general attributes of PSMs is missing, prior analyzing the PSMs effects per type of plant stress. Some potential content information in points (which is not restrictive) is cited below:
- Promotion of plant growth by phosphate solubilizing fungi.
- Information for culturable and non-culturable phosphate-solubilizing microorganisms, emerging solid routes of large-scale availability and microbial applicability.
- Mechanistic quorum sensing (QS) signaling of phosphate-solubilizing bacteria and its potential benefits for agriculture.
- Microbial activity is not independent of plant presence; root exudates stimulate the phosphorus solubilizing ability (as example, bacteria: https://doi.org/10.1038/s41598-023-30915-2). This is a valuable, sustainable insight in using PSMs in agriculture. In lines 294-296 of the text, it is described the effect of PSM to plants without to be mentioned the reverse-way interaction (e.g. the nutritional support of the plant towards PSMs).
- Molecular mechanisms of sensing, transport, signaling and regulating phosphate solubilization in microorganisms (bacteria and fungi); specialized mechanisms for each stress factor could be described at a later stage.
- Physiological, molecular and biochemical differences between phosphate-solubilizing bacteria and/or fungi and how these attributes can affect positively stress alleviation process are not mentioned.
- Direct and indirect modes of for P-solubilization by the microorganisms (Inorganic and organic P-solubilization). Direct and indirect contribution of microorganisms to plant P-uptake.
- Environmental restrictions of using phosphate solubilizing microorganisms; the manuscript is mostly focused on success stories and positive approaches.
- Phosphate-solubilizing microorganisms from air in agriculture (as example https://doi.org/10.3389/fmicb.2020.585215 ).
Response:We sincerely appreciate your insightful suggestions regarding the need for a comprehensive overview of PSM attributes. In direct response to these valuable comments, we have added a new dedicated section (Section 2: "Roles and Multifunctional Traits of PSMs in Soil") that systematically addresses all aspects raised:
(1) General PSM Characteristics:​​
We have added discussion about culturable and non-culturable PSMs (Lines 98-114), and expanded on environmental restrictions and challenges in PSM application (Lines 117-121)
(2) Mechanistic Depth:​​
We have added genetic-level explanations for core processes of P solubilization (Lines 122-150), incorporated quorum sensing regulation of P-solubilization (Lines 143-150), and included plant-PSM bidirectional interactions (Lines 161-180).
(3) Comparative Analysis:​​
We have provided systematic comparison of PSB vs. PSF (Lines 151-160) and distinguished different P-solubilization pathways (Lines 122-143).
(4) Novel Content:​​
We have included refences about PSM from air in agriculture (Lines 106-108) and discussion about plant growth-promoting traits (Line 161-175).
We believe these comprehensive additions have significantly strengthened the manuscript's foundation before delving into stress-specific analyses. We are grateful for these suggestions which have greatly improved our work.
- Citespace analysis did not enclose the term ”plant” and/or “agriculture” for the keyword co-occurrence network. Adding keywords emerges subsections of the review topic which are not discussed. Τhe figure 1 projects literature data analysis from 125 references; this is a very small number of publications for review. There are many research articles (not reviews) with close to 100 references in this journal (https://doi.org/10.3390/plants14091268 with related to the topic abiotic stress). In a broader sense, similar research field reviews e.g. https://doi.org/10.3390/plants14060865 and https://doi.org/10.3390/plants11162119 encloses equal or more than 200 references.
Response:We sincerely appreciate your constructive feedback regarding the CiteSpace analysis and literature coverage. In response, we have made the following substantial improvements:
(1) Expanded Literature Base:​​
We have broadened the Web of Science search strategy with refined keywords, ensuring both comprehensive and accurate coverage of recent research on this topic. We have updated the keyword co-occurrence network (Figure 1) using 261 references (2021–2025), more than doubling the original dataset.
(2)Enhanced Keyword Analysis:​​
We have added a dedicated discussion about core terms (e.g. "plant" and "agriculture") and emerging keyword (e.g., "ACC deaminase"), linking them to mechanistic pathways and agricultural relevance (Lines 73-86).
- Keyword clustering information is missing from the explanation of the figure 1 and/or manuscript text.
Response:We appreciate your comment regarding keyword clustering analysis. In response, we have clarified the search methodology: the Web of Science search query is explicitly provided in the figure legend (Lines 62-66):("phosphate solubilizing" OR "phosphorus solubilizing")AND("microorganism" OR "bacteria" OR "fungi" OR "actinomycetes" OR "archaea")AND("stress tolerance" OR "salinity" OR "drought" OR "heavy metal" OR "pollution" OR "environmental stress")AND ("plant" OR "agriculture" OR "soil"). The corresponding interpretation has been added in Lines 73-86.
- Line 64 “Emerging keywords like "acc deaminase" and "biosorption" suggest a growing interest in mechanistic pathways and remediation processes”: There is no documentation cited if this information is plant- or microbial based since both type of organisms can produce the specific compound (see for bacteria https://doi.org/10.1016/j.micres.2020.126439 ; for plants table 3 https://doi.org/10.3389/fmicb.2015.00937 ).
Response:Thank you for your valuable comment. The production of acc deaminase is microbial based and we have expanded the related discussion in Lines 79-86.
Round 2
Reviewer 2 Report
Comments and Suggestions for Authors
Authors applied all the proposed changes in the manuscript.